# best feature performance in codeswitched hate speech texts

### abstract

How well can hate speech concept be abstracted in order to inform automatic classification in codeswitched texts by machine learning classifiers? We explore different representations and empirically evaluate their predictiveness using both conventional and deep learning algorithms in identifying hate speech in a ~48k human-annotated dataset that contain mixed languages, a phenomenon common among multilingual speakers. This paper espouses a novel approach to handle this challenge by introducing a hierarchical approach that employs Latent Dirichlet Allocation to generate topic models that feed into another high-level feature set that we acronym PDC. PDC groups similar meaning words in word families during the preprocessing stage for supervised learning models. The high-level PDC features generated are based on Ombui et al, (2019) hate speech annotation framework that is informed by the triangular theory of hate (Stanberg,2003). Results obtained from frequency-based models using the PDC feature on the annotated dataset of ~48k short messages comprising of tweets generated during the 2012 and 2017 Kenyan presidential elections indicate an improvement on classification accuracy in identifying hate speech as compared to the baseline

Keywords: *Hate Speech, Code-switching, feature selection, representation learning*

## I. introduction

Identifying hate speech in short text messages retrieved from social media is a challenging classification task. This data is often unstructured and noisy, coupled with the different dimensions of big data which include massive volume, high-velocity, questionable veracity, complex variety, and impactful to society(Burnap & Williams, 2015). The negative ripples of hate speech on social media, easily amplified across time, geographical and legal jurisdictions , not only abrogate user experience but have also been shown to turn into real-life violence in some cases (Hatzipanagos, 2018). This phenomenon is closely associated with online harassment, cyberbullying, body shaming and other hateful attacks targeting individuals or groups on the basis of belonging to a protected characteristic like race, ethnicity, religion, gender, etc.

Social media companies are increasingly under pressure by various stakeholders to better respond to the phenomenon. Evidently, all social media networks have a user policy on hate speech content on their platforms. However, most of these companies mostly rely on users to flag such content which then goes through some element of manual review to ascertain whether it violets the user policy on hate speech content(Sandler, 2018). Subsequently, a decision is made to either to block the content, terminate the account, or take some other predefined action. This approach apparently is unfeasible in the light of big data.

The research community has responded well over time, clearly by the growing number of research activities including workshops, conferences, software applications, and publications in the area of automatic hate speech identification (Warner and Hirschberg,2012; Kwok and Wang, 2013; Gitari et al., 2015), offensive language detection (Xu and Zhu, 2010;) and other related areas like cyberbullying (Cynthia et al.,2015), fake news detection , sentiment analysis, and stance detection. However, most of the studies have focused on European languages, especially English, whereas there are so many other languages used on social media platforms, especially regional and resource-scarce languages like Swahili which is a lingua franca in East and part of central Africa. Besides, monolinguals form about 40% of the world population, whereas bilinguals, multilinguals, and polyglots form about 60%. (Ansaldo et al., 2018) We postulate that this statistics is increasingly being mirrored on social media with the evidence of codeswitched language in datasets, for example the dataset that was built in our study that comprises of ~400k tweets collected during the 2017 presidential campaigns in Kenya. Apparently, majority of Kenyans are multilingual, speaking their mother tongue (L1), Swahili as the national language(L2), and English as the official language(L2) (Muaka, 2011).

Therefore, this study bridges the gap for codeswitched language datasets in regards to automatic hate speech identification. To the best of our knowledge, this is the first study to collect and build a classifier for a codeswitched language dataset, specifically in English, Swahili, Sheng (slang) and s*ome* instances of words from native languages like Gikuyu and Luo. An example text message is,

"*Hawa waarabu wanatorture kenyans wadhani sisi ni madogy zao..... Being our employers doesn't mean u own us .*"
[Translation: "*These Arabs are torturing Kenyans thinking we are their dogs…. Being our employer does not mean you own us.*"]

In general, this study investigates the performance of various features on various machine learning algorithms for hate speech identification in a codeswitched language dataset. Both, conventional and deep learning features and learning algorithms were explored on a ~48k human-annotated dataset using supervised machine learning approach. We actualize the PDC high-level features by mapping them into tf-idf vector representations for the task of hate speech identification. Several experiments using the PDC features prove their effectiveness in capturing hate in short text messages, with Passion (P) and Distance (D) components being the most salient with accuracies of 74.3% and 74.2% respectively. The PDC combined feature set is compared to the conventional features and the human annotation for the same dataset as baseline.

## II. related work

There are several studies that have been conducted in relation to automatic hate speech detection. These can be classified into two main groups: those dealing with general hate speech as a binary classification task and those as a multi-class task. There are more studies in literature dealing with the phenomenon as a binary classification problem whereby the focus is on identifying a specific type of hate speech like racism (Lozano et.al, 2017, Tulkens et al., 2016;Kwok & Wang, 2013), anti-Semitism (Warner & Hirschberg, 2012) or neither. For the second class, the argument is that there is need to differentiate the different types of hate speech or hate speech from offensive messages because conflating them does not match the ground truth. (Davidson et al., 2017;Badjatiya et al., 2017; Fortuna, 2017) .

There are various features that have been used previously in the detection of hate speech in social media text messages including demographic information of the users like the gender, location, (Waseem and Hovy,2016), syntactic features like the length of the message, part-of-speech tags as features and other high-level features, with dictionary lists and part-of-speech tags being popularly used. For the low-level features, the popular feature among the conventional learning algorithms has been the use of word frequency features like Bag-of-Words (BoWs), n-grams (both at character and word-level) and Term Frequency-Inverse Document Frequency (tf-idf) features. Word embeddings as features for deep learning neural network algorithms have increasingly become popular in hate speech detection studies(Badjatiya et al., 2017; Hasanuzzaman et al., 2017; Djuric et al., 2015). The popularly used pre-trained embeddings include Global Vectors (GloVe), FastText n-grams, and Word2Vec text representations, at both character, word and sentence levels. The general feature usage frequency in automatic hate speech studies is well summarized in figure 1.

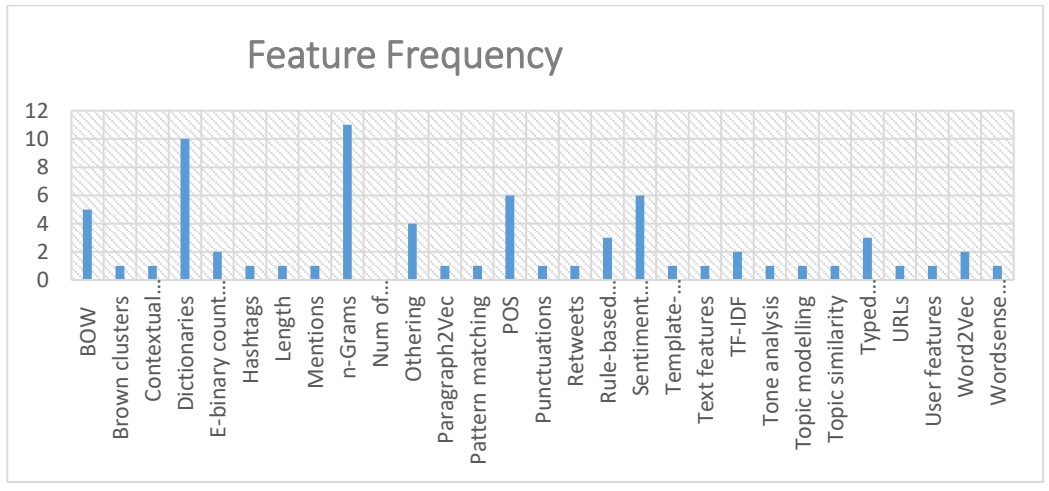

figure 1: Feature usage frequency in literature

Some key observations from studies in literature include the popularity of conventional machine learning algorithm in learning classifiers and that English has been the most popular language of the datasets. According to available literature there very few in other languages like Dutch (Tulkens et al., 2016), Amharic (Mossie & Wang, 2018), Arabic (Al-Hassan & Al-Dosari, 2019) and none in Swahili. In addition, text messages (tweets) from Twitter have been popular with most of these studies. Besides, racism has been the most published type of hate speech. However, hardly any of the studies have publicly shared their full datasets apart from Davidson et al (2017).

## III. method

The general methodology that was used to build the classifier models for hate speech classification in short text messages in this study was based on the CRoss-Industry Standard Processes for Data Mining (CRISP-DM). Therefore, this study's pipeline entailed problem understanding, data acquisition and preparation, feature engineering, modelling, and evaluation.

In order to get a deep understanding of the hate speech phenomenon, a systematic literature review was conducted whereby several related studies and hate theories were examined. In addition, a content analysis was done on various definitions of hate speech as found on user policy guidelines of leading social media networks, and legal definitions of hate speech on constitution documents of various countries, with a focus on the case of Kenya. This was further informed by the need of automating the monitoring for hate speech on social media by key government agencies in Kenya like the National Cohesion and Integration Commision, which is in charge of matters related to hate speech(NCIC, 2019),  the Kenya Education Network  that is the primary Internet Service Provider of all tertiary learning institutions in the Country (Kenet, 2018), and other national security agencies. From this phase, a working definition of hate speech was derived, based on the section 13 of the NCIC Act of 2008 i.e. Hate speech is any message that incites hatred towards a target, whether a person or a group of people, on the basis of belonging to a protected group characteristic like ethnicity, race, religion, etc.

*A.  data*

There were 397,555 raw text messages collected. These mainly consisted of tweets from the 2017 general election in Kenya, including the repeat presidential elections held later the same year.  Additional tweets from the 2012 general elections were also scrapped to form a formidable raw corpus size. This specific period was informed by existing research which indicates that there is often a higher volume of tweets generated during trigger events and not much thereafter (King & Sutton, 2013). Unlike text messages from other social media networks, tweets were purposively chosen because they are often topically structured, publicly available, and programmatically accessible via Twitter APIs, python tweet collection libraries, and even using custom-built crawlers. Moreover, Twitter features that enable users to anonymously engage in public discourse over topical issues in real time makes it a conducive platform for hate speech propagation. Besides, several previous studies in automatic hate speech detection have used tweets (Kwok & Wang, 2013)(Burnap & Williams, 2015)(Waseem & Hovy, 2016).  The main data acquisition strategy employed was bootstrapping, starting with a seed of known hateful keywords(Waseem, 2016) , phrase patterns (William Warner & Hirschberg, 2012), then followed by apparent hateful hashtags during the presidential elections period. In addition, tweets from pro- hate speech Twitter user accounts (Kwok & Wang, 2013), especially bloggers and politicians who often post content bordering hate speech were collected.

Subsequently, this raw dataset was processed and culminated with a clean dataset of kkkk tweets. The data cleaning process included lowercasing, the removal of tweets with unprintable characters, that is the first thirty-two character in the 7-bit ASCII code, removal of non-alphanumeric and duplicate tweets, retweets, emojis, and advertisements that often ride on trending hashtags. Examples of tweets with unprintable characters include "Ã¶ ë¨¼ì¼¼´ì´ì•¼", and with advertisements include  " #NoReformsNoElections Apple launch iPhone X  their most expensive phoneâ€¦" . Moreover, actual mentions were replaced with "USERNAME" tag, and hashtag symbols replaced with "HASHTAG".

 In addition, tweet character length was used. For example below tweets with a length of less than 7 characters were removed. These comprised of tweets with only one or two-characters, with less than 3 word replies, which apparently did not meet the annotation scheme's threshold for the predefined classes of this research. Examples of these include "c", "ok", "DAAMN!", "I will".   For other very short messages, they could not be contextualized because the referenced message was not available or they comprised of only numbers, one-word acronym, or a symbol. Examples of these include "2546","WTF", " Smh..nkt!", "#".  Although some words, whether unigram or bigrams or trigrams, were offensive, they did not have an identifiable target of hate and therefore got dropped too using the criteria of 'less than 4' word length. For example, "F#ck you all!".

In regard to tweet length, the conventional tweet message length has been 140 characters, which now has been doubled to 280 characters. In the contrary, from the observations made in this study, the longest tweet had 991 characters with two long URLs as part of the tweet message. This exception is explained by twitter's design that puts a limit of 280 characters only on the message, whereas anything else thereafter e.g. the URL address can be of any length. Given that the length of a message could probably be an informative classification feature (Davidson et al., 2017) for the study, only the message length was considered, and therefore all the URL part of the tweets were dropped.

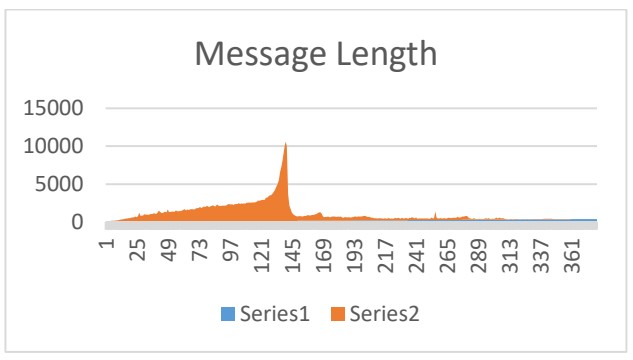

figure 2: Tweet length

*B. data annotation*

Using convenience sampling method, forty human annotators comprising of 80% undergraduate CS students and 20% staff from the school of science and technology at Africa Nazarene University (ANU) were recruited to help in the annotation exercise. The gender was relatively balanced, the average age was 23 but the nationality skewed towards Kenyans. The skewness was informed by the need to have annotators who could easily interpret the codeswitched nature of the collected corpus (mostly in English, Swahili, and some other native languages). An initial training based on the annotation scheme was conducted and the team given sample messages to annotate using a web-based annotation portal developed by the research team (ref). Based on ththe annotation performance and commitment for that specific session, the team was trimmed to 27 annotators. Feedback from the first session was used to  to enhance the annotation portal. A second training session  was conducted and the team contracted to annotate at least 3k messages within a week.  The annotation portal was redesigned to have each tweet annotated by any three random annotators , unlike during the first session whereby each tweet had to be annotated by at least one subject matter expert (SME) and two novice annotators. The new design was infomed by the slow annotation process in the first session and the need to expedite the annotation process in order to have a sizeable labelled dataset for training our classifier.

*C.      Features*

There were four primary features that were used in this study. These are the Bag-of-Words (BoWs), Term Frequency-Inverse Document Frequency (TF-IDF), word embeddings, and the Passion-Distance-Commitment (PDC) features. Other features that were experimented with include Part-of-Speech tags (PoS), and topic models as features. The BoWs features are basically frequency counts of term occurrences in a tweet, were derived using the count vectorizer in Scikit-learn machine learning library. The TF-IDF vectors as features are used to compute the relative importance of a specific term in the tweet and the entire dataset. There were 3 groups of TF-IDF vectors that were derived based on the input token level, that is word, n-grams and character levels. Basically a matrix was generated for the TF-IDF score of each word in the different tweets, another one for the different word combinations (N-grams), and lastly one based on each character in the dataset.

Word Embeddings as features were based on GloVe pre-trained embeddings on the 100d file of about 1 million word vectors.  The tweet dataset was tokenized and each token mapped to their respective embeddings, basically performing transfer learning. Topic Models as high-level features were intentionally used for data exploration and as an automated process to inform the specific words to include in the proceeding phase to generate the PDC family features. Here, Latent Dirichlet Allocation (LDA) was used to generate 23 key topics from the  hate speech dataset. These are as shown figure 3.

PDC features are psycholinguistic features derived from the 3 dimensions of hate as explicated by the triangle of hate theory (Sternberg & Sternberg, 2008). As high-level features, PDC espouses hate speech in 3 primary word-families that are meaning-based and language independents. Therefore, the language list can grow or shrink by adding or removing similar meaning words in different languages in the respective families.  Passion word family consist of words that express emotions of anger, fear, disgust and contempt. These include threatening, abusive, derogatory, and other offensive words directed towards a target person or group that belong to protected characteristics like race, ethnicity, religion, etc. An example message is "to *hell with all <group>. They need to be swept from this country*".  Negative polarity and Sentiment analysis has been used in previous studies to detect passion instances (Chen et al., 2012)(Razavi et al., 2010). Distance word family consists of words that express social distance or proximity in inter-group or persons relationships, what in also referred to as   "othering" language (Burnap & Williams, 2016). This is often indicated by a high frequency usage of pronouns (Semin, 2009) (Coupland, 2010). For example, *"us"," them"," they"," we"," you"*, etc. An example of an actual tweet is "*Kambas also do not make good leaders...they are Cowards*". Commitment word family consists of words or phrases that commit to blatantly hate on another person or group by devaluing. This can either be by referring to  them using object, insect or animal names, or basically perceiving others as less superior, less mature, or less human (Haslam, 2006). Moreover, this includes some of the code names only known and used by the in-group to refer to the members of the out-group. An example tweet is "*Kikuyus Are Enemies of Luos Stop Making Music With This Cockroaches*".

*D. Machine learning classifiers*

Both conventional and deep learning classifiers were used to learn the classifier models. The conventional machine learning algorithms included the Naïve Bayes (NB), Linear Logistic Regression (LLR), Support Vector Machine (SVM), K-Nearest Neighbour (KNN), and Decision Tree (DT). In addition, Bagging and Boosting models, that is Random Forest (RF) and Extreme Gradient Boosting (Xgb) were used respectively.  In regard to deep learning, Convolutional Neural Networks (CNN) and Hierarchical Attention Network (HAN) were explored. There was minimum parameter fine tuning for the deep learning models. All the models used in the experiments were based on Scikit-learn machine learning models.  A good summary of these and their respective classification performance on various features is well summarized on table 1.

*E. Evaluation*

The dataset was split into training and testing datasets: 80% for learning the classifier, and 20% for evaluating the performance of the trained classifier model. There were two key evaluations: evaluation in terms of the inter-annotator reliability score, and the evaluation of how discriminative each feature would be in the text classification task. The Krippendorff's alpha score was used to measure the inter-annotator reliability. Both conventional classifier training algorithms and neural network algorithms were employed to train the text classifier models. The best model was selected using grid search with 5-fold cross validation. Accuracy and F-measure based on precision and recall metrics were used to evaluate the saliency of each feature. For the deep learning algorithms, the iterations were set with an epoch value of 15.

## IV. Results

Topic modeling for the short text messages using the Latent Dirichlet Allocation (LDA) approach derived 23 topics based on the Hate speech class. These are as shown in the 23 rows in figure1. The green cells indicate a legally protected characteristic, in this case the ethnic group names in Kenya and the nationality. The purple cells are individual names, mainly the presidential contenders/politicians and one popular blogger. The blue cells are also groupings but not falling under the protected characteristic category. These include, police, government, country and nation. The yellow cells indicate the "distancing" or "othering" features. The red cells indicate the "passion" features.

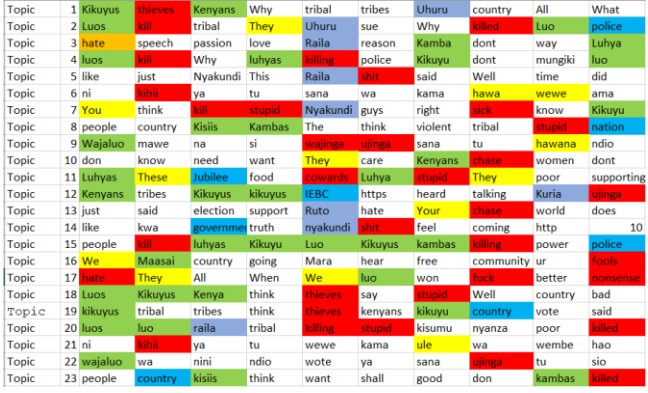

figure 3. Topic modeling for hate speech class

Four primary features were experimented on seven conventional and two deep learning machine learning models. The model performance in terms of accuracy was recorded as shown with the values in table 1. The table rows represent the features whereas the columns represent the respective machine learning algorithms.

Table 1: Feature performance across various ml algorithms

| FEATURE | Machine learning Algorithm | | | | | | | | |
|---|---|---|---|---|---|---|---|---|---|
| Name | Naïve Bayes | Linear LRegression | SVM | KNN | DT | Bagging RF | Boosting Xgb | HAN | CNN |
| 1.BoW | 0.720 | 0.743 | 0.741 | 0.706 | 0.739 | 0.737 | 0.743 | | |
| 2.TF-IDF | | | | | | | | | |
| Word level | 0.747 | 0.752 | 0.741 | 0.732 | 0.738 | 0.735 | 0.741 | | |
| N-gram level | 0.742 | 0.742 | 0.746 | 0.712 | 0.736 | 0.710 | 0.736 | | |
| Character level | 0.739 | 0.751 | | 0.725 | 0.738 | 0.724 | 0.743 | | |
| 3.Word Embeddings | | | | | | | | 0.600 | 0.672 |
| 4.PDC Features | 0.741 | 0.741 | 0.740 | 0.735 | 0.736 | 0.736 | 0.736 | | |

The word embeddings as features were based on the GloVe vector representations (Twitter.27B.100d dataset) with 1193514 word vectors. These were only used for the HAN and CNN models and not on the other models as shown in table 1.

## V. discussion

Topic models as high-level features can be used to inform our PDC high-level features, which in turn can be used to generate informative lower-level features for hate speech classification. For example, from figure 2 in topic 1 (row no.1), "Kikuyus" which is the largest single ethnic group in Kenya is closely linked with "thieves"," Kenyans", "tribal" and "Uhuru". Generally, most of the hate related messages targeting Kikuyus had to do with labelling them as thieves, Kenyans, tribal, and had some relation to one of the presidential candidate "Uhuru".

In topic 2(row no.2), the "Luos", another large ethnic group in Kenya was closely linked with *"kill", "tribal" "Uhuru", "why",* and *"police"*. Hate messages along this topic had to do with questions (*"why"*) of police killings of members of the Luo group during the campaign period. Another interesting topic was no. 9 which included *"Wajaluo"*, which is the Swahili translation for Luo ethnic group. It is closely linked with *"mawe"* (stones), and *"wajinga"* (foolish). A lot of the hate related messages targeting the *"Wajaluo"* were castigating them*," wajinga",* for the violence and property destruction caused as a result of stone-throwing. Generally, the use of psycholinguistic features in the PDC format can further be populated by other terms specified on LIWC (Tausczik & Pennebaker, 2010) to help characterize hate speech text at the high-level.

In regards to general feature performance, the TF-IDF feature outperformed all the other features, particularly at word-level. The deep learning algorithms performance was lower than the conventional models. The lowest performing algorithm was the HAN. Results from the deep learning algorithms could get better with better hyper parameter tuning and more data.

The best algorithm was the linear logistic regression based on TF-IDF word–level features that yielded an accuracy of 75.2%.

Existing tools and techniques in representation learning have mainly focused on one language, with English being the most popular. For example, exiting pre-trained embeddings are mostly in English and other European languages. This becomes a challenge for example when building classifiers to handle hate speech in African and other non-European codeswitched datasets. From the extensive experiments in this study using both conventional, shallow and deep learning algorithms, it is apparent that the conventional algorithms still perform relatively better on smaller datasets as compared to deep learning algorithms. In addition, the existence of codeswitched text negatively impacts the performance of most conventional classifiers. The conventional classifiers are trained to handle text messages, often limited to one language. Therefore, the expected action by the classifier is to drop terms that are incomprehensible, just as it would for most unseen words during training. In this regard, there is need for a better approach to handle codeswitched language datasets.

## VI. conclusion

Linear logistic regression algorithm trained on word-level TF-IDF achieves the best accuracy performance over our codeswitched language dataset. Besides, good results can be achieved inexpensively by using hierarchical structures for high-level features to better inform low-level features selected for training machine learning algorithms. In this study, we have explored the use of topic models as features to inform the PDC feature set which achieved competitive accuracies as shown in table 1. This approach has worked well for a small codeswitched dataset of ~48k tweets and is espoused to perform equally well for multilingual datasets, especially where resource-scarce languages are involved. Future work will involve experimenting with bigger datasets to see how well the approach scales and compare this with conventional features that have been prone to the curse of dimensionality and overfitting.

## Acknowledgment

This study was partly funded by the Kenya Education Network (KENET) under the 2018 CSIS mini-grants research projects.

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
