# OpenReview forum: "Best feature performance in codeswitched hate speech texts"
_ICLR.cc/2020/Conference — Reject_

### Official Review · AnonReviewer1 · 2019-10-06
**Official Blind Review #1**

**Rating:** 3

**Review:**

Comments:

-This paper considers codeswitched hate speech texts from an NLP perspective.  The dataset considers mixed languages.

-Focuses on kenyan presidential election.

-Paper has severe formatting issues as well as simple issues like capitalization.  Additionally many plots are rather unattractive (seem to be produced using Excel or Google Sheets, whereas generally something like matplotlib or seaborn is preferred).

  -The paper puts a lot of examples into the main text, whereas these are usually put into the appendix, or only a few examples are placed in the main text.  Usually the main text focuses more on higher level analysis.

  -Figure 2 is formatted incorrectly (the caption runs on to the next page)

  -I appreciate the effort that went into data annotation as well as the disclosure of the demographics of annotators.

  -Table 1 should make the metric much clearer (it's mentioned in the main text, but it should be in the caption too, also the best performance usually should be bolded)!  Generally TF-idf features or PDC features seem to have the best performance.  The performance of the CNN does not seem very strong.  I think a simple RNN based approach might also be worth considering.  It would also be worth analyzing if the differences between the methods is attributable to underfitting or overfitting.

  -If the paper is proposing a new task with many baselines, it's also important to release the dataset and code in my opinion (I believe ICLR allows this to be done in a de-anonymized way).

Review:

This paper deals with an important problem in social media analysis.  With the spread of hate speech and hate crimes by rioting separatists in Hong Kong as well as equally hateful attacks on Chinese people in the west, I think that this is an issue that deserves more attention in our community.  Unfortunately this paper needs more polish to be appropriate for ICLR.  It also might be better suited to an NLP focused conference (such as ACL, EMNLP, or NAACL) although I think if the technical contribution is clear enough it could be suitable for ICLR as well.

I think the big things to focus on would be including more baselines, improving polish in the paper, and providing a clearer high-level analysis of the dataset (with specific examples mostly left for the appendix).

**Experience Assessment:**

I have published one or two papers in this area.

**Review Assessment: Checking Correctness Of Derivations And Theory:**

N/A

**Review Assessment: Checking Correctness Of Experiments:**

I assessed the sensibility of the experiments.

**Review Assessment: Thoroughness In Paper Reading:**

I read the paper at least twice and used my best judgement in assessing the paper.

---

### Official Review · AnonReviewer2 · 2019-10-22
**Official Blind Review #2**

**Rating:** 1

**Review:**

This paper compared several classification methods, including deep neural networks (DNN), to identify hate speech texts. Mainly the data was corrected from twitter. The data was prepossessed to deal with by popular classification methods. The LDA and PDA are used to construct the identifier with high accuracy. Finally, Practical data was used to assess the performance of several machine learning algorithms.

The research topic is important from the sociological viewpoint. I'm not sure whether this paper suits to the publication from ICRL. Besides that, the authors did not show any technical insight into the numerical results. That is, can the authors explain the reason why the linear logistic regression with TF-IDF features outperformed all the other methods? Overall, this paper did not provide any useful knowledge, while this paper introduced some statistical methods and showed numerical results. I recommend the authors to add more beneficial insight and to submit the paper to other conferences that deals with sociological issues.

**Experience Assessment:**

I do not know much about this area.

**Review Assessment: Checking Correctness Of Derivations And Theory:**

N/A

**Review Assessment: Checking Correctness Of Experiments:**

I did not assess the experiments.

**Review Assessment: Thoroughness In Paper Reading:**

I made a quick assessment of this paper.

---

### Official Review · AnonReviewer3 · 2019-10-22
**Official Blind Review #3**

**Rating:** 1

**Review:**

I'm sorry to say that this paper is not ready for publication.

I think it's an important area and the dataset collected could be quite valuable for tackling hate speech.

The paper does not follow the style guide, is full of typos or 'kkkkk' tokens indicating missing values. The first sentence of the abstract is not grammatical. Codeswitched needs to be mentioned more specifically in the introduction.

I couldn't easily find statistics about the dataset, especially in terms of language breakdown. How many of the tweets were multi-lingual? For pure-english tweets, I would be interested in this dataset being split out as a sub-dataset, as I would heavily bet the sota method for classifying hate-speech would be to fine-tune a BERT model on these labels. We need a table of dataset statistics.

We need a mathematical definition of PDC that is made very explicit in the paper, there is too much prose, I did not have time to do the background reading of the linked papers to understand this sociological theory of hate speech.

The paper did not feel sufficiently anonymized. I would anonymize the university used to create the dataset and the funding agencies that supported the research in subsequent submissions.


**Experience Assessment:**

I have published one or two papers in this area.

**Review Assessment: Checking Correctness Of Derivations And Theory:**

I carefully checked the derivations and theory.

**Review Assessment: Checking Correctness Of Experiments:**

I assessed the sensibility of the experiments.

**Review Assessment: Thoroughness In Paper Reading:**

I read the paper at least twice and used my best judgement in assessing the paper.

---

### Decision · Program_Chairs · 2019-12-19

**Decision:**

Reject

**Comment:**

This paper focuses on hate speech detection and compares several classification methods including Naive Bayes, SVM, KNN, CNN, and many others. The most valuable contribution of this work is a dataset of ~400,000 tweets from 2017 Kenyan general election, although it is unclear whether the authors plan to release the dataset in the future.

The paper is difficult to follow, uses an incorrect ICLR format, and is full of typos.

All three reviewers agree that while this paper deals with an important topic in social media analysis, it is not ready for publication in its current state. The authors did not provide a rebuttal to reviewers' concerns.

I recommend rejecting this paper for ICLR.